# The Potential of the Synthetic Strigolactone Analogue GR24 for the Maintenance of Photosynthesis and Yield in Winter Wheat under Drought: Investigations on the Mechanisms of Action and Delivery Modes

**DOI:** 10.3390/plants10061223

**Published:** 2021-06-16

**Authors:** Mojde Sedaghat, Yahya Emam, Ali Mokhtassi-Bidgoli, Saeid Hazrati, Claudio Lovisolo, Ivan Visentin, Francesca Cardinale, Zeinolabedin Tahmasebi-Sarvestani

**Affiliations:** 1Department of Environmental and Plant Biology, Ohio University, Athens, OH 45701, USA; 2Department of Plant Production and Genetics, School of Agriculture, Shiraz University, Shiraz 7144165186, Iran; Yaemam@shirazu.ac.ir; 3Department of Agronomy, Faculty of Agriculture, Tarbiat Modares University, Tehran 14115111, Iran; mokhtassi@modares.ac.ir (A.M.-B.); tahmaseb@modares.ac.ir (Z.T.-S.); 4Department of Agronomy, Faculty of Agriculture, Azarbaijan Shahid Madani University, Tabriz 53714161, Iran; saeid.hazrati@azaruniv.ac.ir; 5Department of Agricultural, Forest and Food Sciences (DISAFA), University of Turin, 10095 Grugliasco, Italy; claudio.lovisolo@unito.it (C.L.); ivan.visentin@unito.it (I.V.); francesca.cardinale@unito.it (F.C.)

**Keywords:** drought, foliar application, GR24, irrigation, strigolactone, wheat

## Abstract

Strigolactones (SLs) have been implicated in many plant biological and physiological processes, including the responses to abiotic stresses such as drought, in concert with other phytohormones. While it is now clear that exogenous SLs may help plants to survive in harsh environmental condition, the best, most effective protocols for treatment have not been defined yet, and the mechanisms of action are far from being fully understood. In the set of experiments reported here, we contrasted two application methods for treatment with a synthetic analog of SL, GR24. A number of morphometric, physiological and biochemical parameters were measured following foliar application of GR24 or application in the residual irrigation water in winter wheat plants under irrigated and drought stress conditions. Depending on the concentration and the method of GR24 application, differentiated photosynthesis and transpiration rate, stomatal conductance, leaf water potential, antioxidant enzyme activities and yield in drought conditions were observed. We present evidence that different methods of GR24 application led to increased photosynthesis and yield under stress by a combination of drought tolerance and escape factors, which should be considered for future research exploring the potential of this new family of bioactive molecules for practical applications.

## 1. Introduction

Strigolactones (SLs) are a structurally diverse class of carotenoid-derived signaling molecules. For about four decades, SLs have been known as root exudate components and germination stimulants for seeds of parasitic plants, an obviously detrimental role to the producing plant. In 2005, they were proven to boost hyphal branching in arbuscular mycorrhizal fungi, thus stimulating symbiosis establishment. Since 2008, they have been recognized as a family of phytohormones, for which the first assigned endogenous role was in the inhibition of shoot branching/tillering (for a comprehensive review, see [1]). Since then, many additional effects have been added, such as the promotion of secondary shoot growth and leaf senescence, the inhibition of adventitious rooting [2] or the stimulation of lateral root density and epidermal cell length [3], as well as crop yield and seed quality [4]. Furthermore, it is now becoming apparent that SLs contribute to long-term survival in various harsh environmental conditions [5]. For example, as mentioned above, when roots produce and release SLs into the soil as they do especially under nutrient-limiting conditions, they promote the symbiosis with beneficial fungi [6] and thus, indirectly improve their own nutrition [7]. It is also suggested that SLs are important for direct acclimation responses and plant plasticity in response to nutrient availability, notably by inhibiting branching/tillering and shaping root architecture under low nutrients [8].

During drought acclimation, a complex crosstalk between SLs and ABA (abscisic acid) has been highlighted [5], with important differences depending on the organ (root vs shoot) and on the type of plant (monocot vs dicot). In rice, SLs seem to inhibit ABA synthesis, and thus, most SL-related mutants are reported as drought resistant [9]; however, it is not known what the short- or long-term effects of exogenous SLs—such as the synthetic analog GR24—would be on stomatal conductance or stress responses and general performance of monocots; only in winter wheat, initial studies from our group and others have highlighted a promotion of drought tolerance by GR24 treatment [10,11]. In dicots, the picture seems more nuanced. SL production in tomato (*Solanum lycopersicum* L.) and *Lotus japonicus* roots decreases, possibly to allow locally for the physiological increase of ABA [12]. In shoots, all data converge instead in showing that SLs play a positive role in drought stress tolerance and avoidance [13,14]. SL-depleted *Arabidopsis thaliana* (Arabidopsis), tomato and *L. japonicus* exhibit higher sensitivity to drought stress in comparison to wild-type plants [13,15,16]. In Arabidopsis, the application of GR24 rescues the drought sensitive phenotype of SL-depleted mutants and increases the drought tolerance of wild-type plants over the level of the untreated control [13]. In tomato and Arabidopsis, GR24 induces a transient stomatal closure that was proven, in the latter, to be independent of ABA [14,15]; on the other hand, endogenous SLs are clearly able to increase stomatal sensitivity to and/or movement of ABA in leaf tissues of dicots [12,13,14,16].

SL signaling has been also associated with reactive oxygen species (ROS) responses in plants [17]. In a previous research on winter wheat (*Triticum aestivum* L.) plants, foliar application of GR24 has been shown to increase antioxidant enzymes activity and to decrease malondialdehyde, which are signs of a higher plant potential for drought tolerance; however, no information on the deployment of stress avoidance mechanisms (e.g., stomatal closure) or escape (e.g., higher biomass allocation to the roots) are available in wheat or other monocots [10].

Thus, because of their intriguing biological properties, SLs are attracting interest not only in basic plant biology research but also for their great potential for applications in agriculture. They indeed can be seen as tools to manipulate shoot and root architecture, to stimulate root colonization by arbuscular mycorrhizal fungi and enhance plant nutrition; they have been also proposed as tools to manage drought resilience [5,18]. In this regard, understanding their effects on plant physiology and biochemistry, along with the practicalities of their application on different crops under various conditions relevant for commercial production, may lead both to insights into how plants cope with harsh environmental conditions such as drought, as well as to agricultural innovations. In this manuscript, we focused on trials in which the effectiveness of the synthetic SL analog GR24 to improve winter wheat performances under drought was investigated. Namely, two possible application methods for GR24 were compared: by leaf spraying or by delivery via irrigation water. In parallel, the effects of such treatments on physiological and biochemical parameters important for crop performances under stress were investigated. The results provide further insights into the direct effect of SLs on the antioxidant machinery in a monocot plant and suggest potential translational avenues for the application of SL research in agriculture.

## 2. Results and Discussion

### 2.1. Plant/Atmosphere Gas Exchanges Are Altered by GR24 Treatments

Data collected in this work about photosynthesis rates indicated that both GR24 application methods led to a less strong reduction in this parameter under drought stress (Figure 1a); for a comparison with the values in irrigated samples, see Appendix A Appendix A. The effect was more marked in plants that had received GR24 in the residual irrigation water than as foliar spray (*p* ≤ 0.05). So, GR24 treatment mitigates the detrimental effect of drought on photosynthesis.

Photosynthesis rates strongly depend on stomatal conductance, and stomatal conductance correlates negatively with stomatal limitation and CO_2_ availability to fixation [19,20,21]. Thus, it was of interest to determine whether the application of GR24 affects not only photosynthesis but also these other parameters. Gas exchange data indicated that mock-treated, drought-stressed plants (Figure 1b) had higher stomatal limitation compared to unstressed controls (Appendix A Appendix A) but also that GR24 could bring stomatal limitation down both if delivered via the residual irrigation water and—though less pronounced—via foliar application (*p* ≤ 0.05) (Figure 1b; both under stress). The stomatal limitation in stressed plants treated with 10 µM GR24 in the residual irrigation water was close to the unstressed control values (Appendix A Appendix A). Consistently, stress reduced stomatal conductance and transpiration rates (Appendix A Appendix A) but less so in GR24-treated than mock-treated plants; the effect was stronger if GR24 had been delivered via the residual irrigation water (Figure 1c,d).

Leaf water potential is considered a measure of the water status of a plant [22]. Thus, leaf water potential values were compared in unstressed controls (Appendix A Appendix A) and stressed plants (Figure 1e) that had been mock treated or had received GR24 via the residual irrigation water or foliar application. The data showed that in stressed plants, leaf water potential values were less negative in GR24-treated plants with either method than in mock-treated plants. However, this change was more pronounced in plants treated via foliar application (Figure 1e). Thus, while stomatal limitation was higher (and stomatal conductance lower) upon foliar than irrigation application of GR24, leaf water potential was less negative after the former than the latter treatment. This led to an apparent contradiction between the higher stomatal conductance and less negative water potential values in stressed plants treated with GR24 vs untreated. To try and explain it, we estimated via the LI-COR software the intercellular (substomatal) CO_2_ concentration of leaves (C_i_), which is a critical parameter in photosynthesis and stomatal conductance. The results showed that in stressed and GR24-treated plants, C_i_ values decreased in comparison with equally stressed but mock-treated plants, and that the decline was sharper when GR24 had been added to the residual irrigation water (Figure 1e). It is noteworthy that also in unstressed controls, photosynthetic rates in GR24-treated plants showed an increasing trend that became statistically detectable at 10 µM (Appendix A Appendix A).

As highlighted in the introduction, SLs have been demonstrated to affect stomatal closure with different outcomes in monocots and dicots. While data in dicots largely support the hypothesis that SLs promote stomatal closure in ABA-dependent and independent ways [13,15] and excess SLs leads to better performances under stress also via higher sensitivity to ABA [4,23], a different pattern has been suggested in monocots. In rice leaves for example, ABA synthesis appears to be repressed by endogenous SLs, and stomatal limitation is decreased under osmotic stress in response to GR24 [9]. It must be noted here that the application of GR24 in the absence of drought stress did not significantly alter stomatal conductance, measured 48 h after treatment (see Appendix A Appendix A). It is possible that if a transient, direct effect on stomata was induced under non-stressing conditions, it was already undetectable 2 days after treatment, as reported for other plants. Alternatively, SLs may have a similar antagonistic relationship with ABA in wheat as they have in rice, and the higher stomatal conductance and transpiration in our GR24-treated wheat leaves may be due to a decrease in endogenous ABA. This is an issue of high importance that would deserve further, dedicated investigation; however, the less negative water potential in GR24-treated plants vs untreated argues against this hypothesis. On the other hand, the higher stomatal conductance in GR24-treated plants upon drought stress could be driven by the lower C_i_ values. Hence, our results also imply that the application of GR24 could alleviate the drop in photosynthetic performances due to drought stress [23] in wheat, and that this, in turn, tends to keep stomata open to keep up the CO_2_ supply.

A lower need to deploy stress avoidance strategies such as stomatal closure may be due to increased tolerance via better strain mitigation or stress escape via better water capture, and these strategies may not be necessarily deployed to the same extent in different species or genotypes within the same species. For example, Zivcak et al. (2008) [24] have compared several winter wheats genotypes in drought conditions, and they have found that the genotypes differed in stomatal conductance, and hence in the CO_2_ assimilation rate. Genotypes with higher drought tolerance also display a delayed stomatal closure and higher stomatal conductance caused mainly by higher osmotic adjustment; this leads to higher net assimilation rate and higher production of assimilates and consequently higher yield. The ability to maintain turgor under water deprivation in spite of a relatively sustained gas exchange rates is what we observe especially for plants to which GR24 was delivered with the residual irrigation water. It should be noted here that stomatal conductance can be affected by osmolyte adjustment or changes in temperature, as proved in van ‘t Hoff equation. In our previous experiments with GR24, we could indeed observe higher proline concentrations in winter wheat plants that had received GR24 [10]. However, whether or not osmotic adjustment is different in GR24-treated plants, water availability should be increased in the leaves of the plants that were treated with GR24 in the irrigation water, to justify the more negative water potential than leaf-sprayed plants combined with higher stomatal conductance.

### 2.2. GR24 Treatment Increases Root Biomass, Root-to-Shoot Ratio and Yield in Winter Wheat

One of the most parsimonious hypotheses to be explored in order to explain a higher water availability in GR24-treated plants, especially in the group treated via the residual irrigation water, could be more performant roots that are better at water capture. This could be achieved via a relatively larger root apparatus and/or lower tissue resistance to water flow, for example by increased aquaporin activity. While we have no direct molecular indication on the latter hypothesis in our experimental system, aquaporin genes have been found to be dysregulated in SL-related Arabidopsis mutants undergoing drought [13], making this point worthy of investigation. Concerning the former hypothesis, the root and shoot dry weight (g per plant) of all treatments were measured in both irrigated and drought stress conditions in our experiment. The results showed higher root weight in GR24-treated plants, with root dry weight greater in plants that had received GR24 with the irrigation water (Figure 2a), and the same trend was true for shoots (Figure 2b). Moreover, the extent of weight gain was higher in roots than shoots of plants that had received GR24 in the residual irrigation water, thus increasing their root-to-shoot ratio (Figure 2c). Note that a similar pattern is visible also in the absence of stress (Appendix A Appendix A). GR24-treated plants had also a higher yield than untreated controls (Figure 2d, Appendix A).

Thus, the skewed resource allocation towards the roots and the higher water availability this entails for the shoots, coupled to the lower C_i_ values reported in Figure 1e for the GR24-treated plants (especially if in their residual irrigation water), may contribute to the marked water-spending behavior displayed by the latter group, which kept their stomata more open than untreated controls irrespective of their water potential.

### 2.3. GR24 Treatment Decreases Hydrogen Peroxide (H_2_O_2_) Content

As a byproduct of oxidative aerobic metabolism, H_2_O_2_ is continuously produced in plants under stress [25]. Additionally, one of the adverse consequences of any stress, including drought, is an accumulation of cellular ROS, which will be converted to H_2_O_2_ in the enzymatic scavenging process. H_2_O_2_, as a ROS, functions in signal transduction pathways and gene expression modulation in plants under abiotic stresses [26]. The H_2_O_2_ content in wheat plants was significantly decreased by the application of GR24 under drought stress, compared to mock-treated plants (*p* ≤ 0.05). The lowest H_2_O_2_ content was observed in plants that received 10 µM GR24 via the irrigation water (Figure 3); a similar, though less marked pattern, was visible in unstressed control plants treated with the highest GR24 concentration (Appendix A Appendix A). As it is known that H_2_O_2_ in guard cells promotes ABA-regulated stomatal closure [27], lower H_2_O_2_ in GR24-treated leaves could be (in principle, and if reflected in guard cells) a contributing reason for higher stomatal conductance.

### 2.4. Antioxidant Enzymes Activity

Drought stress or other environmental challenges induce metabolic imbalances that can cause oxidative stress in plant cells. To cope with such stress, plants usually rely on antioxidant defenses, which can be enzymatic or non-enzymatic. The former are usually considered as the most effective; the major enzymatic categories involved are superoxide dismutase (SOD), ascorbate peroxidase (APX), peroxidase (POX), and catalase (CAT). The latter two rapidly get rid of excess H_2_O_2_ presumably to allow low steady-state levels in pathways of redox signaling. CAT is considered as an essential element for the removal of H_2_O_2_ produced in the peroxisomes during photorespiration [28]. Furthermore, SOD catalyzes the superoxide radical conversion to H_2_O_2_, and POX uses H_2_O_2_ for substrate oxidation in the cytosol, vacuole and cell walls [29]. Moreover, in wheat plants, it has been proposed that all environmental cues including drought [30] induce oxidative damage, emphasizing the importance of regulating the antioxidant system efficiently to cope with these abiotic stresses. In this regard, the maintenance of high levels of antioxidants is considered beneficial for the plant, which thereby becomes able to counter the negative effects of ROS [30]. In our experiments, we observed an overall clear trend towards the induction of antioxidant enzymatic activities in leaves of GR24-treated plants under drought stress. The results are shown in Figure 4: application of GR24 with the residual irrigation water significantly enhanced the activities of antioxidant enzymes in all treatments (*p* ≤ 0.05), with the 10 µM concentration being the most effective. On the other hand, foliar application of GR24 induced a significant difference in comparison with mock-treated, drought-stressed plants for all measured antioxidant activities, except APX, only at the higher concentration (10 µM). As expected, a negative correlation between H_2_O_2_ content (Figure 3) and antioxidant enzymes activities was observed, especially the H_2_O_2_-scavenging CAT and APX. These results show that the application of GR24, especially via the roots, might help wheat plants to attain higher drought mitigation and tolerance in comparison to mock-treated plants, thanks to potentiated antioxidant mechanisms. The induction of the antioxidant machinery by GR24 has been suggested earlier [10,31]. As discussed in the previous paragraphs, this—together with the morphological plasticity demonstrated in Figure 2—might also diminish the need for stress avoidance via stomatal closure, which can explain why we observed higher stomatal conductance in response to GR24 in this experiment.

Once again, GR24 effects may be achieved via crosstalk with other hormones or a more direct effect on the antioxidant system. For example, a modulation of ABA levels or sensitivity by GR24 may promote osmotic adjustment via proline [32] and aquaporin expression [13,33] but also induction of antioxidant defenses and, as a consequence, suppression of ROS-driven damage [30]. It is worthy of note that in tomato, leaves with high levels of endogenous SLs will have lower stomatal conductance under no or mild stress conditions compared to leaves with wild-type levels of SLs, due to higher ABA sensitivity of their guard cells; however, the former will be able to keep significantly higher stomatal conductance levels and photosynthesis rates under drought stress than the latter, likely due to better drought tolerance and less need to deploy avoidance [12]. However, we did not measure plant hormone concentrations in this work, so while a certain degree of ABA involvement in the observed phenotype is likely, it remains speculative at this stage. As a final note, the racemic mixture of GR24 used here contains equimolar amounts of two enantiomers, which were shown in Arabidopsis to activate both the SL pathway and the sibling pathway initiated by KARRIKIN INSENSITIVE 2 (KAI2) [34], also involved in drought resistance [35]. Therefore, even though this unexpected bioactivity in the KAI2 pathway was not proven to occur in other plant species yet, care should be exerted in ascribing the results to either pathway whenever rac-GR24 is used.

## 3. Conclusions

Given the effects of SLs in several biological systems, scientists are now working to find practical applications for these molecules to improve crop performances [5]. GR24 is a stable chemical compound with interesting features towards microorganisms in the rhizosphere, besides an undeniable promoting effect on plants stress tolerance; this gives it great potential in agricultural applications. In this work, we compared two GR24 delivery methods on a commodity crop, confirming that GR24 could boost winter wheat tolerance and escape to drought, and showing that the delivery method could differentially affect the intensity and efficacy of the physiological and biochemical responses to drought stress. Indeed, while these results confirm the positive effect of GR24 in drought irrespective of the delivery way, some features of the response were specific to the method, offering suggestions to translate basic plant biology insights into fine-tuned field applications.

## 4. Materials and Methods

### 4.1. Plant Materials and Growth Conditions

This study was conducted in the greenhouse of the College of Agriculture, Shiraz University, Shiraz, Iran. Seeds of *Triticum aestivum* L. cv. Sirvan (relatively drought-tolerant) [36] and an extensively cultivated variety in Iran) were provided by the Seed and Plant Improvement Institute, Karaj, Iran. *Rac*-GR24 (GR24 hereby) was kindly supplied by Professor Binne Zwanenburg (Radboud University, Nijmegen, NL). The experiment was evaluated using a completely randomized design with four replications. Wheat seeds were germinated in a growth chamber (28/20 °C day/night) on filter paper and then sown in 7–liter plastic pots. The pots were filled with soil:sand in a 2:1 ratio (soil classification: fine, mixed, mesic, Cacixerollic Xerochrepts). According to the soil analysis at the time of planting, the final electrical conductivity, pH, and available N, P, and K of the experimental soil were 0.60 dS m^–1^, 7.09, 0.15%, 12 mg kg^–1^ and 720 mg kg^–1^, respectively. All pots were fully irrigated (100% of field capacity, FC) for twenty days after sowing; only unstressed controls were kept at 100% FC during the whole experiment. Data from these irrigated plants are provided in the supplementary material. Besides mock-treated and unstressed controls, five treatments were carried out with four plants in each pot and 4 pots, for a total of 16 plants/treatment pooled in 4 replicates/treatment: drought stress with foliar application of GR24 (5 and 10 µM), drought stress with application of GR24 (5 and 10 µM) via the residual irrigation water, drought stress and mock treatment (see below). For plants under water-deficit stress, soil moisture content was maintained at 40 ± 5% FC. Soil water content in each pot was measured using a TRIME-FM TDR (Time Domain Reflectometry, IMKO Micromodultechnik, Ettlingen, Germany). Water stress treatments were initiated 20 days after planting, at which time the plants had 3 expanded leaves. GR24 was applied twice for both the foliar application method and the application with irrigation: at the tillering (after 14 days of drought stress treatment) and anthesis stages (after 60 days of drought stress treatment). The GR24 stock solution was dissolved in acetone and then distilled water was added to reach the desired dilution, while the control plants were mock-treated with a water and acetone solution corresponding to the 10 µM GR24 treatment. For both groups, treatment was done via foliar spraying or in the residual irrigation water. To ensure GR24 uptake in plants treated by foliar application, leaves were sprayed completely and homogeneously until runoff. Samples for biochemical analyses were collected 2 days after the last GR24 treatment, i.e., from 82-day-old plants after 62 days of drought treatment. At the same time, whole plants were harvested, and morphological features were measured. Shoots including spikes were removed from the soil surface, soil was carefully washed from the roots, and different plant parts were dried separately for a week at 65 °C before weighing. Based on the root and shoot dry weight (g per plant), root/shoot ratios were calculated.

### 4.2. Hydrogen Peroxide Content

Hydrogen peroxide (H_2_O_2_) content was determined spectrophotometrically according to a published method [37]. H_2_O_2_ was extracted by homogenizing 0.5 g leaf samples in 0.5 mL of 0.1% (*w*/*v*) trichloroacetic acid (TCA). The homogenate was centrifuged at 12,000× *g* and 4 °C for 10 min. The reaction mixture contained 0.5 mL of leaf extract supernatant, 2 mL reagent (1 M KI in double-distilled water) and 0.5 mL of 10 mM K-phosphate buffer (pH 7.0). The blank probe consisted of 0.1% TCA in the absence of leaf extract. The reaction was let develop for 1 h in darkness. The amount of H_2_O_2_ was calculated using a standard curve prepared with known concentrations of H_2_O_2_ (measured by absorbance at 390 nm), and the H_2_O_2_ content was expressed as µmol g^−1^ of fresh weight (FW).

### 4.3. Preparation of Enzyme Extract and Antioxidant Enzymes Assays

Superoxide dismutase (SOD, EC 1.15. 1.1) activity was measured according to a published method with some modifications [38]. Flag leaves (500 mg FW) were homogenized in 5 mL extraction buffer consisting of 50 mM Na-phosphate buffer (pH 7.8), 0.05% (*w*/*v*) β-mercaptoethanol and 0.1% (*w*/*v*) ascorbate. For the assay, 3 mL of 50 mM Na-phosphate buffer (pH 7.8) containing 9.9 mM L-methionine, 57 μM nitro-blue tetrazolium (NBT) and 0.0044% (*w*/*v*) riboflavin were mixed with 0.2 mL of enzyme extract. The reaction was terminated after 10 min by removing the test tubes from the light source. Purple formazan, the reaction product of NBT, was measured at 560 nm. The supernatant volume corresponding to 50% inhibition of the reaction in this assay was assigned a value of 1 enzyme unit.

For the assays of catalase (CAT, EC 1.11.1.6), ascorbate peroxidase (APX, EC 1.11.1.11) and peroxidase (POX; EC 1.11.1.7), we used fully expanded flag leaves (500 mg FW) immediately frozen in liquid nitrogen and homogenized by a homogenizer in 1 mL of ice-cold 0.1 M K-phosphate buffer (pH 7.8) containing 1 mM ethylenediamine tetraacetic acid (EDTA), 1 mM ascorbic acid, 0.5% (*v*/*v*) Triton X-100 and 2% (*w*/*v*) polyvinylpyrrolidone (PVP K-12). Insoluble materials were removed by centrifuging at 12,000× *g* for 20 min at 4 °C, and the supernatants were used for the enzymatic assays as described below.

The CAT activity was measured by monitoring the disappearance of H_2_O_2_ at 240 nm in a reaction mixture consisting of 0.3 mL of 100 mM H_2_O_2_, 1.5 mL of 50 mM Na-phosphate buffer (pH 7.8) and 0.2 mL of leaf extract [39]. The activity was reported as enzyme units (μmol of H_2_O_2_ decomposed per minute) per FW gram of leaf (ε = 39.4 mM cm^−1^).

The APX activity was measured on 0.2 mL of leaf extracts as described [40], added to 1.0 mL reaction mixture containing 0.1 mM EDTA, 0.1 mM H_2_O_2_, 0.5 mM ascorbate and 50 mM K-phosphate buffer (pH 7.0). The H_2_O_2_-dependent oxidation of ascorbate was followed by decreased absorbance at 290 nm (ε = 2.8 mM cm^−1^). One unit of APX was defined as the amount of enzyme that breaks down 1 μmol of ascorbate min^−1^ g^−1^ of protein.

The POX activity was assayed using 0.2 mL of leaf extract mixed with 1.78 mL of reaction mixture containing a 50 mM phosphate buffer (pH 7.0) and 0.05% guaiacol, followed by the addition of 20 µL of 10 mM H_2_O_2_. The oxidation of guaiacol was measured by the increase in absorbance at 436 nm (ε = 26.6 mM^−1^ cm^−1^) for 1 min in the presence of H_2_O_2_ [41]. One unit of POX was defined as the amount of enzyme that caused an increase in the absorbance of 0.01 min^−1^.

### 4.4. Measurement of Leaf Water Potential and Gas Exchange Rates

The leaf water potential was measured in all plant groups using a pressure chamber technique (PMS instrument company, ALBANY Oregon 97322) 48 h after GR24 application at anthesis in the GR24-treated groups along with net photosynthetic rate, stomatal conductance, transpiration rate, stomatal limitation and substomatal CO_2_ concentration (C_i_) on fully expanded flag leaves in four plants per group using an infrared gas analyzer (LICOR, Lincoln, NE 68504, USA) during daytime between 10:00 a.m. and 2:00 p.m., with photosynthetic photon flux density exceeding 1800 µmol m^−2^ s^−1^. Stomatal limitation data were presented as percentage, calculated according to the following equation: stomatal limitation = (1 − C_i_/C_a_) × 100, where C_a_ is the ambient CO_2_ concentration.

### 4.5. Statistical Analysis

The experiments were repeated twice, and the collected data were subjected to the analysis of variance (one-way ANOVA) using the SAS statistical software package SAS 9.1. When appropriate, means were compared using the LSD test (*p* ≤ 0.05). All the values are expressed as the means of four replicates ± standard error (SE).

## Figures and Tables

**Figure 1 plants-10-01223-f001:**
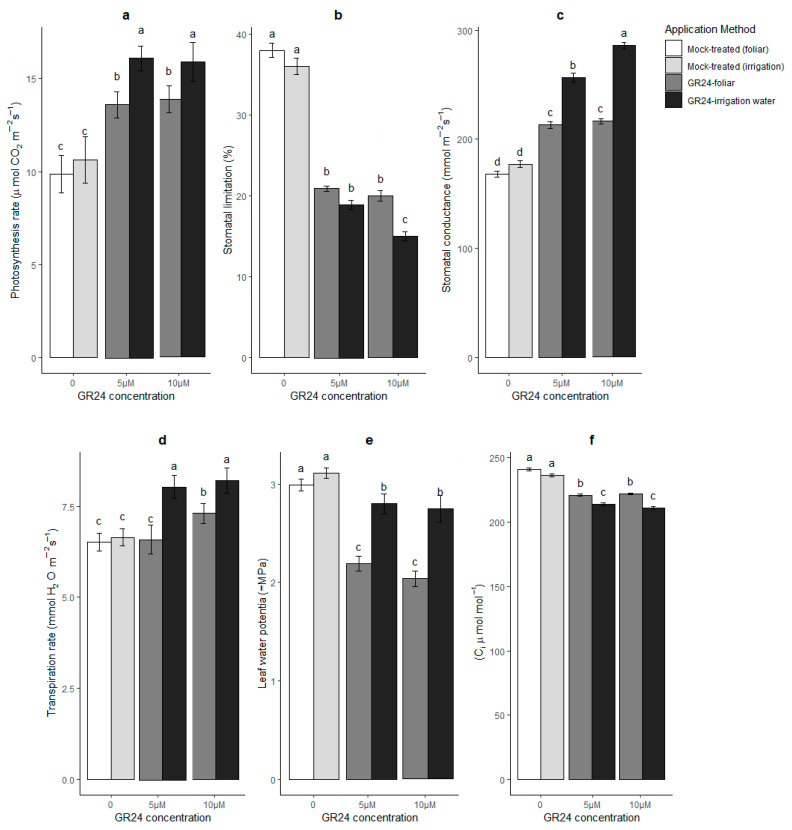
Photosynthesis rate (**a**), stomatal limitation (**b**), stomatal conductance (**c**), transpiration rate (**d**), leaf water potential (**e**) and substomatal CO_2_ levels (C_i_) (**f**) in wheat plants in response to two methods of GR24 application under drought stress conditions. Mock-treated plants in these graphs were equally drought-stressed but received a water and acetone solution; for irrigated control values, see Appendix A Appendix A. Each value represents the mean ± SE (*n* = 4, each replicate the pool of four plants). Different letters on top of bars indicate significantly different means for *p* ≤ 0.05 (LSD test).

**Figure 2 plants-10-01223-f002:**
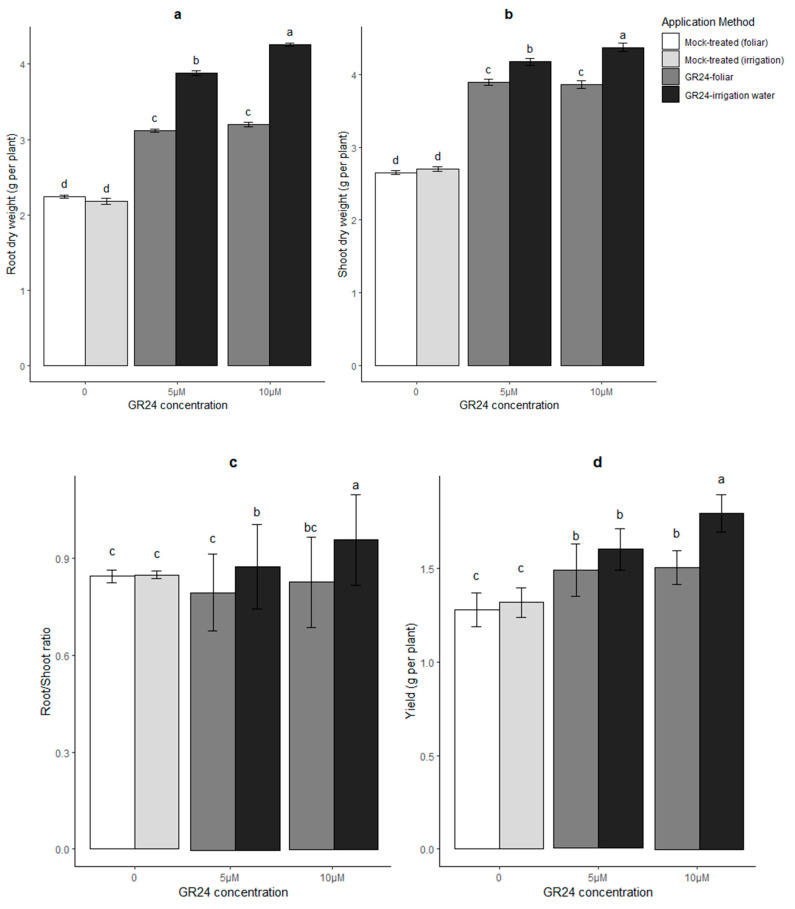
Root dry weight (**a**), shoot dry weight (**b**), root/shoot ratio (**c**) and yield (**d**) in wheat plants in response to two methods of GR24 application under drought stress conditions. Mock-treated plants in these graphs were equally drought-stressed but received a water and acetone solution; for irrigated control values, see Appendix A Appendix A. Each value represents the mean ± SE (*n* = 4, each replicate the pool of four plants). Different letters on top of bars indicate significantly different means for *p ≤* 0.05 (LSD test).

**Figure 3 plants-10-01223-f003:**
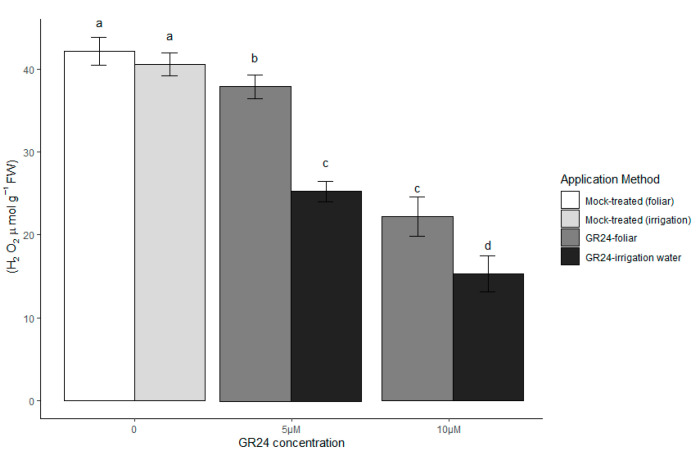
H_2_O_2_ in wheat plants in response to two methods of GR24 application under drought stress conditions. Mock-treated plants in these graphs were equally drought-stressed but received a water and acetone solution; for irrigated control values, see Appendix A Appendix A. Each value represents the mean ± SE (*n* = 4, each replicate the pool of four plants). Different letters on top of bars indicate significantly different means for *p* ≤ 0.05 (LSD test).

**Figure 4 plants-10-01223-f004:**
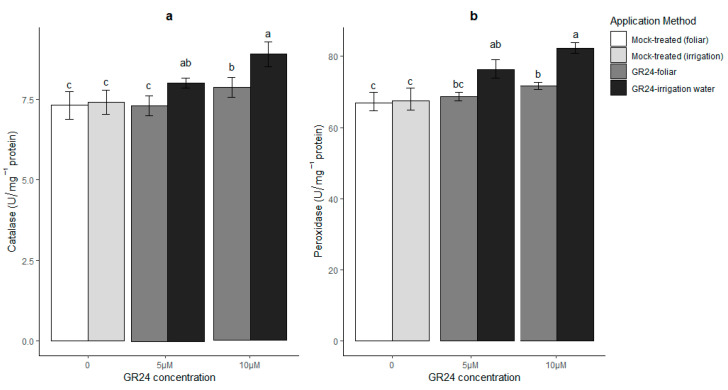
Catalase (**a**), peroxidase (**b**), superoxide dismutase (**c**) and ascorbate peroxidase (**d**) enzyme activities in wheat leaves under drought stress conditions, in response to two application methods for GR24. Mock-treated plants in these graphs were equally drought-stressed but received a water and acetone solution; for irrigated control values; see Appendix A Appendix A. Each value represents the mean ± SE (*n* = 4, each replicate the pool of four plants). Different letters on top of bars indicate significantly different means for *p* ≤ 0.05 (LSD test).

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
