# Peer review of "The Potential of the Synthetic Strigolactone Analogue GR24 for the Maintenance of Photosynthesis and Yield in Winter Wheat under Drought: Investigations on the Mechanisms of Action and Delivery Modes"

_plants, 2021, doi:10.3390/plants10061223_

Round 1
Reviewer 1 Report
It is likely that the authors addressed the points raised up the reviewers, and the manuscript has improved well. There are still some issues that the authors should address before its publication. The details are described as below.
1, Abstract line 20; ‘a synthetic analog of SLs and SL-like molecule named GR24’
⇒’a synthetic analog of SL, GR24’ should be correct?
2, P3 line99-111; The authors are comparing some results in Figure 1 and supplementary Figure 1. But the Y-axis scales are different between two figures, and it would be difficult to find the difference between these two figures. Probably it would be better not to separate these two figures.
3, The authors are using a word ‘pants that received GR24’. I think this explanation is a bit strange. It would better to say ‘the plants that were treated with GR24’.
4, P7, line 291; ‘enzymatic on non-enzymatic’ ⇒ ‘enzymatic or non-enzymatic’?
Author Response
Dear Reviewer,
On behalf of all co-authors, I would like to thank you for your time spent critically reviewing the manuscript and providing the valuable comments that have enhanced this paper.
Yours sincerely

Reviewer 2 Report
I read the paper “The Potential of the Synthetic Strigolactone Analogue GR24 for the Maintenance of Photosynthesis and Yield in Winter Wheat under Drought: Investigations on the Mechanisms of Action and Delivery Modes” submitted for re-review with interest. The authors responded and satisfactorily explained all the issues raised by me in the previous review. The work is written much better and easier for the reader. My only remark is that the summary should emphasize the specific result of the research carried out in line with the purpose of the work, e.g. depending on the GR24 concentration and method used of application, differentiated photosynthesis efficiency in drought conditions was observed/not observed, the method of application had/did not have a significant impact on tested parameters e.t.c. I also agree with the authors that it would be difficult for them to add results for other varieties and I understand the choice of this particular variety for research, but in the future I propose to conduct research on a wider material.
Author Response

(The authors gave the same response as above.)

Reviewer 3 Report
Authors made extensive revisions according to reviewers' recommends. This MS is in this form is highly applicable for publication in this journal. This MS can contribute to further studies of strigolactones during stress responses.
Author Response

(The authors gave the same response as above.)
